# Do customers' perceptions of Islamic banking services predict satisfaction and word of mouth? Evidence from Islamic banks in Bangladesh

Muhammad Khalilur Rahman[1,2], Muhammad Nazmul Hoque[3]*, Sharifah Norzehan Syed Yusuf[4]*, Mohd Nor Hakimin Bin Yusoff[1], Farhana Begum[5]

1 Faculty of Entrepreneurship and Business, Universiti Malaysia Kelantan, Kota Bharu, Malaysia, 2 Angkasa-UMK Research Academy, Kota Bharu, Malaysia, 3 Faculty of Accountancy, Universiti Teknologi MARA Cawangan Selangor, Bandar Puncak Alam, Selangor, Malaysia, 4 Accounting Research Institute, Universiti Teknologi MARA, Shah Alam, Malaysia, 5 Islamic Development Bank (IsDB), Ministry of Education, Dhaka, Bangladesh

* nazmul@uitm.edu.my (MNH); shari893@uitm.edu.my (SNSY)

**Data Availability Statement:** All relevant data are within the paper and its Supporting Information files.

## Abstract

This study aims to investigate the customers' perceptions of Islamic banking services and their impact on satisfaction and word of mouth (WOM) with others. This study designs the bootstrapping procedures using a partial least square method to test path coefficient results. Structured questionnaires were distributed among clients of Islamic banks in Dhaka city, where 377 responses were collected for data analysis. The findings revealed that there is a highly significant relationship between security and customers' perception. Ethical responsibility and religious value have a positive and significant impact on customers' perception whereas benefit has a negative significant impact on customers' perception. Findings from this study also indicated that customers' perceptions mediate the effect of ethical responsibility, religious value, benefit, and security on satisfaction. In addition, customers' satisfaction mediates the effect of customers' perception and WOM. These findings can promote managers of Islamic banks to build customer satisfaction and WOM with Islamic banking services, and attain competitive advantage that may lead Islamic banks to succeed in the competitive business. This study also provides new insights into customers' WOM with others about Islamic banking services. This knowledge could assist Islamic banks to understand the customers' perceptions that would increase satisfaction and in turn, contribute to WOM with others in determining where would be best to target marketing attention of Islamic banking services with limited resources.

## Introduction

Conventional banks practice that money generates money with interest-based transactions. From an Islamic point of view, money can produce money if it is utilized in a marketable

**Funding:** The authors would like to thank the support from Accounting Research Institute HICoE of Universiti Teknologi MARA for research grant funding and Malaysian Ministry of Education. The funders had no role in study design, data collection and analysis, decision to publish, or preparation of the manuscript.

**Competing interests:** The authors have declared that no competing interests exist.

transaction, where a basic service is used. The holy book of Al-Quran has forbidden any financial dealings that contain *Riba* (interest). Islamic banking not only prohibits interest, but the industry also prohibits its players from engaging with gambling activities, speculation, alcohol, and other unethical activities [1]. Most Islamic banking products are established from current mainstream banking products by removing any forbidden elements, making them Shariah-compliant [2]. Therefore, Islamic financial institutions must confirm that their customers are satisfied with their products, and believe in customers' perception towards Islamic banking services. As such, Islamic financial institutions all over the world have continuously strived to improve their banking products and services to satisfy potential customer perceptions and demands [3, 4].

However, principles that are related to Islamic banking products and services, such as ethical responsibility, religious values, customer benefits, and security issues, are facing challenges to meet with customers' perception and their satisfaction, particularly from the perspective of Bangladesh. These challenges are mostly prevalent among practitioners and Shari'ah council members, as well as society at large because they are potential users of Islamic banking products and services [4]. In an assessment of these challenges, this study investigates the Islamic banking customers' perceptions with their satisfaction and word of mouth (WOM) for Islamic banking services. By improving the quality of services, and taking into consideration ethical responsibility practices as well as security, Islamic financial institutions may achieve customer satisfaction. This in turn can reflect customers' WOM with others by motivating new customers to start using Islamic banking products and services. Fida et al. [5] also indicated that bank customers are fascinated with Islamic banks that offer simple and convenient banking facilities. In addition to that, customers are also attracted to Islamic banks that take responsibilities and action to increase customers' satisfaction. Resources of Islamic banks have been assessed to be US$1.5 trillion in 2018, and it was estimated that these resources would achieve as much as 3.2 trillion in 2020 [5]. These resources offer added choice to a huge number of individuals who would benefit from the Islamic banking industry.

In 2017, the Islamic Financial Services Board (IFSB) indicated that the worldwide Islamic financial sector records for a significant proportion of money with a 79 percent contribution to the total Islamic finance industry [5]. This represents the significance of Islamic banking sector in the overall Islamic finance ecosystem. Islamic banking sector in Bangladesh has also been growing significantly, thanks to the well-formed strategy aided by Bangladesh Bank and public interest parties. Bangladesh Bank has issued a sovereign investment Sukuk which will smooth liquidity administration of Islamic banks, and it supports financial deficit and encourages Islamic capital sectors to increase assets for foundation and industrial developments towards accomplishing higher comprehensive growth in gross domestic product (GDP) with sustainable development goals (SDGs). By the end of 2020, eight full-fledged Islamic banks in Bangladesh worked with 1311 branches out of all 10752 branches of the entire financing industry. Furthermore, there are 19 Islamic financial sectors of nine mainstream banks, where 198 Islamic banking windows of 14 mainstream banks are additionally offering Islamic banking services in Bangladesh [6] Between October and December of 2020, deposits and investments increased by 2.28 percent and 3.55 percent accordingly. In the meantime, remittance and additional liquidity of Islamic financing grew by 19.24 percent and 60.61 percent in comparison with that of the last quarter [6]. Therefore, it ensures that the Islamic banking industry has been contributing vastly to Bangladesh's economic sectors.

In this study, customers' perception of Islamic banking services are measured by understanding factors that would predict customers' perceptions of Islamic banking services. Asdullah and Yazdifar [7] classified the factors in accordance to their individual, cultural, and social features. Several other studies found factors such as reputation, trust, cost and benefits, service

quality, satisfaction, attitude, and loyalty intention to be considered by individuals when choosing Islamic banking services [8–11]. Ethical responsibility is also an important determinant of Islamic finance. This is because banks need to disclose information on goods that are connected to their services, products, and fair trading to business places [8, 12], which are profoundly connected to ethical organizations. Directed with Islamic law and guidelines for all the financial transactions in Islamic financing, employees need to understand ethical responsibility that would create customer satisfaction. Religious value is another important determinant towards customers' perception of Islamic banking services as the nature of these services are heavily referred to religious obligations from Islamic Shariah-based business aspect. Newaz et al. [13] identified that there is a direct relationship between religiosity, attitude, and intention towards Islamic banking services. Thus, it is important to examine how religious value influences customers' perceptions of Islamic banking services.

Benefits is also a significant factor in the Islamic financing industry. It is useful to examine whether Islamic investments can be capable of offering benefits in Islamic financial services [14]. Empirical findings previously showed that the expansion of benefits developed religious value investment, and some determined that religious value investment is capable to offer various benefits for foreign investors [15–17]. These findings encouraged this study to explore more recent empirical evidence on the issue of benefits that predicts consumers' perceptions of using Islamic banking services. Security issues in Islamic financing industry are one of the crucial determinants. Zeshan [18] observed that customers would generally choose Islamic banking services because they provide convenient processes, safety, security, and rapid delivery of services. Some scholars realized that consumers' perceptions of Islamic banking services are influenced by security. As Islamic finance continues to ensure strong security, more customers would intend to be involved in Islamic banking services [19]. Based on the discussion above, this study aims to investigate factors that influence customers' perceptions of Islamic banking services. Further, this study tests the impact of customers' perception on satisfaction and also WOM, which is influenced by satisfaction. This study includes a theoretical foundation as well as the development of the study hypotheses with predicting consumers' perception of Islamic banking services within the context of Bangladesh. Subsequently, this study discusses on the research methodology, research findings, discussion, and concluding remarks.

## Literature review

### Underpinning theory

This study has employed the idea of expectation-confirmation theory to measure customers' satisfaction and their WOM about Islamic banking services. In 1980, Oliver developed the expectation-confirmation theory which aimed at the customers' perception of satisfaction. This study extends to measuring the customers' word of mouth (WOM) using the concept of expectation confirmation. The theory also attempts to justify customers' satisfaction and customers' internal factors for using Islamic banking services. Several models have been established within the Islamic banking literature to determine customers' psychological behavior toward Islamic banking products and services. Rahman et al. [20] justified that tourists' satisfaction has a significant and positive impact on WOM for travel destinations. Other studies also found that customers' satisfaction played an important role in influencing WOM. For instance, Han et al. [21] showed that halal goods and products contributed a significant role in raising tourists' satisfaction, which impacts WOM for traveling tourism spots. Similarly, Battour et al. [22] identified that satisfaction and trip value affect WOM. However, from an Islamic banking perspective, customers' satisfaction is crucial for buying products and services which leads to WOM for further recommendation in selecting Islamic banking services. Harris

and Khatami [23] pointed out that satisfaction, trust, quality, commitment, and perceived value are significant predictors of WOM. Mahadin and Akroush [24] found that perceived value, convenience, and satisfaction are identified to be the key factors of WOM. Based on the concept of confirmation or disconfirmation theory, this study examines factors (e.g., ethical responsibility, religious value, benefits, security) that reflect customers' perception of Islamic banking service and their satisfaction, which predicts WOM with others in further consideration of Islamic banking services.

## Ethical responsibility

In recent years, ethics is one of the important determinants in Islamic banking sectors due to the emphasis on having good connections among customers. Previous studies showed that Islamic financial institutions have been taking responsibility to reveal evidence regarding their products, services, and better trade-in marketplace [12]. These are profoundly related to ethical responsibility. Based on Islamic law, employees that are heavily involved with operating transactions in Islamic banks should understand their ethical responsibility and commitment, which would ultimately develop customers' satisfaction in their services. Hadi and Muwazir [25] investigated that mostly the Malay community would care about ethical issues when they select banking services. However, this is not really the case for those customers from the Chinese and Indian communities. They further suggested that Islamic banks should implement religious and ethical issues, as well as develop good service quality. Similarly, Ezeh and Nkamnebe [26] suggested that Islamic banks should improve ethical standards in order to achieve more customers' satisfaction. Based on these discussions, this study proposed that:

H1: Ethical responsibility has a significant influence on customers' perception of Islamic banking services.

## Religious value

The importance of religious value or religiosity is one of the factors that motivate customers to use Islamic financing services and perceptions of Islamic banking products. This provides individual willingness to utilize Islamic banking services. Bananuka et al. [27] defined "religiosity as the religious commitment that denotes participation in, or endorsement of practices, beliefs, attitudes or sentiments that are associated with an organized community of faith." There are several studies that have found positive and significant relationships between religious values and customers' perception of Islamic banking services. Kaawaase and Nalukwago [28] investigated that there is a direct effect between religiosity and customers' perception of Islamic banking services in Uganda. Likewise, there is also a significant direct effect between religiosity and customers' perception of Islamic banking services [29]. Religiosity, perception, and attitude have positive links to different Islamic banking services. Thus, this study formulated the following hypothesis:

H2: Religious value has a significant influence on customers' perception of Islamic banking services.

## Benefits

Customers' benefits in bank services refer to the financial practices and operations that are risky and usually kept away by Islamic financing companies. By performing careful audits and investigations, Islamic finance promotes the decrease of risk and makes the space for greater investment stability [30]. Justice and fairness are crucial components of the fundamental

Islamic banking model that depends on the benefit-sharing principle, by which the risk is shared by the bank and customer [31]. This arrangement of financial intermediation contributes to a more impartial distribution of income and wealth. Islamic banks offer credit at favorable terms and conditions, lower service charges, and monthly repayment which are more profitable than conventional financing. Fusva et al. [32] indicated that Islamic banks offer economical and service-derived benefits that are superior to those of other banks. In the Islamic financial industry, benefits is an important factor. It is useful to examine whether Islamic investments can be capable of offering benefits in Islamic financial services. Numerous countries have realized the benefits of services and products provided by Islamic banks. For instance, Souiden and Rani [33] mentioned that the government should open its business market for both national and international Islamic finance and increase people's consequences of the benefits of Islamic banking. Similarly, Saiti and Noordin [34] indicated that local investors can achieve additional benefits by expanding their investments in other capital markets. Dusuki and Abdullah [35] mentioned that the benefit of the products and services given by Islamic finance is one of the significant issues to develop a customer's perception to utilize Islamic financing services. Therefore, this study proposed that:

H3: Benefits have a significant impact on customers' perception of Islamic banking services.

## Security

Security is a crucial dimension for measuring customers' perceived value on banking transactions because security concerns are connected with technology-based services [36]. It is common for everyone that the banking industry is using Information Technology (IT) in fulfilling organizational activities, as well as meeting bank customers' demands for financial transaction activities. In a financial context, security is a financial instrument that has monetary value and can be traded. Security issues are important in Islamic banking systems, which may affect customers' perception towards Islamic banking services. Arcand et al. [37] (2017) defined customer-perceived security to be involved within transmission of information through digital technology. Poon [38] identified that security plays an important role in determining bank customers' acceptance of internet banking services. Some banking activities are done every day, such as commercial transactions, transfer funds, ATM, credit cards, and banking loans. Banks should protect customers' personal information, therefore they should not disclose customers' private information to other parties without authorization. Islamic banks should ensure constant updates of high-security patches are being put in place for banking transactions. This is because high security of banking transactions would positively influence customers' satisfaction. Zeshan [18] identified that consumers would choose Islamic banking because of its convenient procedure, security, safety, and speedy services. Butt et al. [39] indicated that in the Islamic banking service, security and safety are major factors for consumers to select Islamic finance. This study investigates the relationship between security and the perception of customers toward satisfaction with Islamic banking services. Lai [40] highlighted the importance of the banking transaction security issue and the Islamic financial system. Thus, this formulated that:

H4: Security has a significant influence on customers' perception of Islamic banking services.

## Customers' perceptions

Customers' perceptions of Islamic banking service is a very important component concerned with customers' experience and satisfaction. Therefore, this study is carried out to examine the

relationship between customers' perceptions of Islamic banking services and satisfaction. Singh and Kaur [41] mentioned that if customers expect a certain amount of a service where they feel that facilities have offered more than what they assumed, then they will be satisfied. On the other hand, if the facilities are poor then their expectation will be dissatisfied. Several researchers have identified the relationship between customers' perception and satisfaction with banking services. For instance, Dhurup et al. [42] found that different factors of customers' perception have a positive significant effect on satisfaction with banking services. Anouze et al. [43] identified that customers' perception of Islamic banking in Jordan has a direct link to customers' satisfaction. Most recently, a study conducted by Mahajan [44] found a positive significant effect between customers' perception and satisfaction with online banking services. Based on this empirical evidence, this study has proposed:

H5: Customers' perception has a significant impact on their satisfaction with Islamic banking services.

## Satisfaction and word of mouth (WOM)

WOM in this study refers to bank customers' interest in banking-related financial products and services, which is reflected in their day-to-day discourse. Satisfaction refers to the act or pleasure of fulfilling a need and expectation of a customer towards products and services. Bank customers' satisfaction and positive WOM have become the main objectives focused on by directors or managers of the banks [45]. In the banking industry, the effect of customers' satisfaction on WOM has been playing a significant role, as customers are expecting better quality from services and products [46]. Customers' satisfaction is deemed as one of the crucial components for justifying WOM or customer loyalty. Hence, (WOM) is a form of customer loyalty to products and services. Additionally, Siddiqi [47] implied that satisfaction is an important factor in the well-competitive finance sector. It is because the ability to keep a long-standing connection with its consumers is worthwhile [48]. Previously, there were other researches that identified the relationship between customers' satisfaction and WOM. For example, customers' satisfaction has influenced WOM positively [49]. Abdul-Rahman et al. [50] found that the role of customers' satisfaction affects WOM significantly. In Islamic banking services, especially Islamic credit cards, customers' satisfaction has a strong connection to WOM. Based on this discussion, this study intends to examine the direct relationship between customers' satisfaction and WOM. Thus, this study have postulated that:

H6: Satisfaction has a significant impact on word of mouth towards Islamic banking services.

Based on the review of literature and underpinning theory, this study have formulated the research model (Fig 1).

## Materials and methods

This study was conducted with customers of Islamic banks in Bangladesh. This study measures customers' opinions about Islamic banking services, which includes customers' perceptions (ethical responsibility, religious value, benefits, and security), satisfaction, and word of mouth (WOM) with others. In this study, ethical responsibility is associated with the good connection between employees of Islamic banks and customers. This association is measured using four items modified from Hadi and Muwazir [25], also Ezeh and Nkamnebe [26]. To measure the religious value of customers' views about Islamic banking services, three items were adapted from Kaawaase and Nalukwago [28], as well as Yasin et al. [29]; whereas four items were modified from Dusuki and Abdullah [35] to evaluate the benefit of Islamic banking services. Four

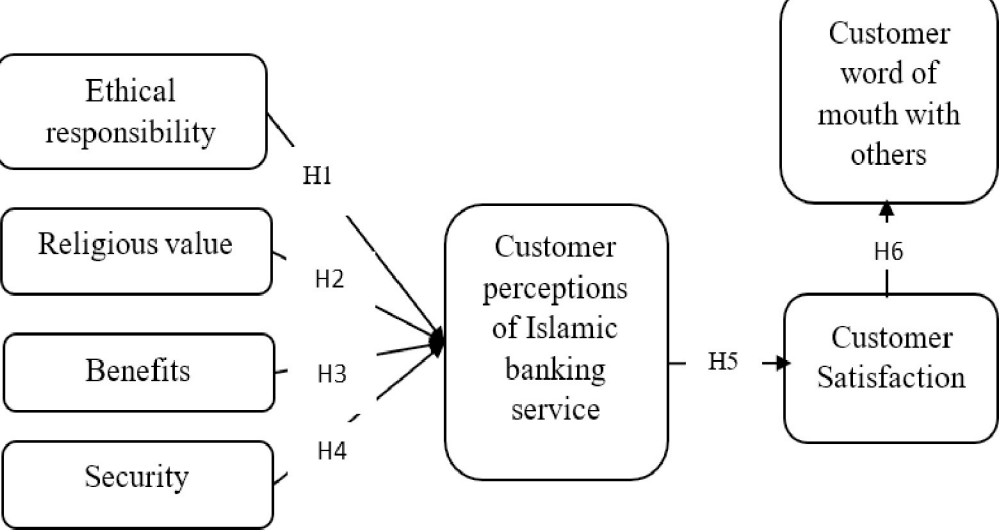

**Fig 1. Research model: Author's elaboration.**

items were modified from Zeshan [18] and Butt et al. [39] to measure the security issues of Islamic banking services. To evaluate customers' perception of Islamic banking services, four items were adapted from Anouze et al. [43] and Mahajan [44]. A total of ten items were modified from Al-Msallam [49] and Abdul-Rahman et al. [50] to measure customers' satisfaction and their WOM about Islamic banking services. Customers' level of satisfaction or dissatisfaction about their perceptions, satisfaction and word of mouth about Islamic banking services in Bangladesh were measured using the five-point Likert scale from 1 (strongly disagree) to 5 (strongly agree).

This study employed a survey method and designed a self-administered questionnaire following the scale adapted from literature review. The questionnaire was evaluated by multiple subject and language experts in the area of Islamic banking products and services before data collection. Prior to data collection, this study had also conducted a pilot test to test the reliability of the questionnaire. Johanson and Brooks [51] postulated that 30 samples are enough to be used in a pilot study for evaluating the reliability of the questionnaire. The pilot test was conducted using Cronbach's alpha. Cronbach's alpha measures the internal consistency between items on a scale. A total of 50 questionnaires were distributed for the pilot test to respondents from various Islamic banks in Dhaka City. From these, 30 completed questionnaires were returned, providing a 60 percent response rate. Data analysis of the pilot test was done using Statistical Package for Social Science (SPSS) software version 22.0. Responses from the completed questionnaire were tested using Cronbach's alpha for the reliability of the questionnaire. Findings of Cronbach's alpha value ranged between 0.781 and 0.869. In the pilot test, a few interesting comments and suggestions were obtained from respondents. For instance, a few respondents suggested adding an option for "don't know/not sure or not applicable" on the Likert scale. Some respondents also commented that statements for some question items were more or less similar to one another. This had given confusion for the respondents to give proper response. Taking into account these remarks, the questionnaire was hence rechecked and amended before data collection resumes.

Suggestions from experts and respondents' opinions (similar meaning of questions, ambiguity, and biases) were incorporated in the finalized version of the questionnaire so that all question items are easy to be understood by respondents. Data was then collected from clients

of Islamic banks in Dhaka city, which is the capital city of Bangladesh in terms of business, financial, and cultural aspects. This study used a judgment sampling and transaction interception method, in which clients of Islamic banks were approached to participate in the survey when they were leaving after completing any banking transaction. This survey method is similar to Hidayat et al. [11, 52], which has also touched on the context of Islamic banking aspects. The anonymity of participants in this study is maintained and considered unbiased as this study ensures that their responses are completely voluntary and will be used only for academic purposes.

In addition, according to the rules of the ethics committee of Universiti Teknologi MARA, an ethical approval letter was obtained to collect data from customers. There was no risk involved and this study had no intention to publish anyone's personal information. This study was conducted in accordance with the Declaration of Helsinki. Informed consent for participation was obtained from respondents who participated in the survey. An introductory letter with consent form was attached to the questionnaire, and respondents were requested to read the ethical statement before proceeding to provide their responses. Respondents were assured that there was no compensation for this survey, and their information will remain confidential.

This study had physically distributed 700 questionnaires to bank customers of Islamic banks in Bangladesh between the period of April and July 2021. 385 responses were returned from bank customers, where eight questions were found to be incomplete. After the initial screening process, this study identified a total of 377 valid responses for data analysis (e.g., excluding missing data, and elimination outlines), which resulted in a 53.86 percent response rate of data collection. Approximately 315 questionnaires were not returned from respondents because they were preoccupied with their transactions and left without returning the questionnaires. Bank customers were waiting for the call number of their transaction purposes when they received the questionnaire. As they were busy with their transactions, it became more difficult for this study to get a high rate of response from bank customers. Thus, a 53.86 percent response rate deemed to be enough for this study. The findings revealed a proportion of 76.1 percent male and 23.9 percent female respondents. Details of the respondents are reported in Table 1.

## Results and analysis

### Common method bias and normality analysis

In this study, common method bias (CMB) was measured using partial least squares method. If the collinearity statistics (VIF) value is greater than 3.3, it is an indication of pathological collinearity that affects a model common method bias. According to Kock [53], if all values of VIF from the collinearity test are equal to or lower than 3.3, the model can be considered to be free of common method bias. In addition, to assess the normality test of data, this study had considered skewness and kurtosis tests. Sheridan and Coakes [54] recommended that skewness value should be between -1.5 and 1.5 whereas kurtosis value is between -2.0 and 2.0 to be within the range of normality. The results identified the skewness and kurtosis value for each variable to be acceptable, which signifies that the data is normally distributed. Mean and standard deviation scores are reported in Table 2.

### Measurement model analysis

The analysis of the measurement model evaluates the outer model of the proposed research model and explains the convergent and discriminant validity of this study. The evaluation of measurement shows the relatedness of measurement items from respective latent variables.

**Table 1. Demographic information.**

| Characteristics | Number | % | Characteristics | Number | % |
|---|---|---|---|---|---|
| Gender | | | Marital status | | |
| Male | 287 | 76.1 | Married | 190 | 50.4 |
| Female | 90 | 23.9 | Single | 185 | 49.1 |
| Total | 377 | 100.0 | Others | 2 | .5 |
| Age | | | Total | 377 | 100.0 |
| 20–30 years old | 220 | 58.4 | Education | | |
| 31–40 years old | 89 | 23.6 | Secondary School certificate | 29 | 7.7 |
| 41–50 years old | 55 | 14.6 | Higher Secondary/Equivalent | 42 | 11.1 |
| 50+ | 13 | 3.4 | Bachelor/Equivalent degree | 154 | 40.8 |
| Total | 377 | 100.0 | Master/Equivalent degree | 145 | 38.5 |
| Monthly income | | | PhD | 7 | 1.9 |
| Below USD200 | 92 | 24.4 | Total | 377 | 100.0 |
| More than USD600 | 43 | 11.4 | Job status | | |
| USD201-300 | 106 | 28.1 | Others | 10 | 2.7 |
| USD301-400 | 73 | 19.4 | Public employee | 22 | 5.8 |
| USD4001-500 | 35 | 9.3 | Private employee | 254 | 67.4 |
| USD501-600 | 28 | 7.4 | Student | 91 | 24.1 |
| Total | 377 | 100.0 | Total | 377 | 100.0 |

The model is estimated using factor loadings, which is required to have an acceptable value that is greater than 0.70 (Table 2) for the respective constructs [55]. Average variance extracted (AVE) results should be higher than 0.50 [55], whereas Cronbach's alpha (CA), rho_A, and composite reliability (CR) should be greater than 0.70, respectively [56]. Table 3 summarized the results of convergent validity for latent constructs.

This study has assessed the discriminant validity of this study using Fornell-Larcker criterion [57] (Fornell & Larcker, 1981) and Heterotrait-Monotrait (HTMT) ratio [58]. Fornell-Larcker criterion states the exceeding of the square root of average variance extracted whereas comparing it with the latent variable correlations. According to the criteria of HTMT ratio, the HTMT ratio should not be greater than 0.85. Table 4 reports the meeting of the requirement of both Fornell-Larcker criterion and HTMT ratio requirement. In addition, for the robustness of discriminant validly, this study reports the cross-loading results of each construct. The findings considered discriminant validity because all item loadings exceeded higher scores on their respective constructs (Table 5). Hair et al. [59] reported that all indicators require high loading score on its construct and low loading on the other constructs.

## Structural model analysis

After achieving requirements of the measurement model, this study proceeds to evaluate the structural model with hypothesis testing. The predictive relevance and accuracy of the model are measured by R-square that recognizes coefficient, and q-square that reflects cross-validated redundancy using blindfolding analysis. According to Cohen [60] and Hair et al. [55], the r-square score should exceed 0.26 and the q-square score above 0 (zero) is termed as substantial (Table 6). The results revealed that most of the hypothesis relationships are positive and significant at p-value 0.01 and 0.05 levels except for the negative significant impact between benefits and perceptions (β = -0.088, t-value = 1.741) at 0.05 level of significance. The highest impact between security and perception (β = 0.532, t-value = 7.690), perception and satisfaction (β = 0.809, t-value = 36.331), and (β = 0.785, t-value = 24.407) are found to be positive and

**Table 2. Reliability analysis.**

| Characteristics | Mean | SD | Skewness | Kurtosis | VIF | FL |
|---|---|---|---|---|---|---|
| Ethical responsibility | | | | | | |
| Islamic banks have an ethical responsibility for dealing the customers (ER1) | 4.08 | 1.260 | -1.208 | 0.281 | 2.087 | 0.841 |
| I like ethical banking service (ER2) | 4.04 | 1.209 | -1.136 | 0.296 | 1.854 | 0.814 |
| Islamic bank provides equal rights to each customer (ER3) | 3.77 | 1.193 | -0.687 | -0.316 | 2.150 | 0.827 |
| Islamic bank does not invest in unlawful activities (ER4) | 4.29 | 1.076 | -1.563 | 1.752 | 2.338 | 0.871 |
| Religious value | | | | | | |
| Islamic financial institutions and agencies that provide Islamic financing are compliant with Islamic law (RV1) | 4.08 | 1.096 | -1.066 | 0.374 | 1.940 | 0.832 |
| Religion motivates me to seek Islamic financing (RV2) | 4.25 | 1.125 | -1.480 | 1.337 | 1.492 | 0.768 |
| Islamic bank upholds the Islamic image and provide the social welfare (i.e. giving donations or scholarships) (RV3) | 4.04 | 1.183 | -1.069 | 0.221 | 2.235 | 0.838 |
| Benefit | | | | | | |
| Islamic banking is more profitable than conventional financing (BF1) | 3.77 | 1.273 | -0.691 | -0.610 | 1.708 | 0.790 |
| Islamic banking offers credit at favorable terms and conditions (BF2) | 4.02 | 1.107 | -0.974 | 0.224 | 1.803 | 0.840 |
| Islamic banking offers lower service charge and lower monthly repayment (BF3) | 3.91 | 1.152 | -0.788 | -0.266 | 2.539 | 0.857 |
| The costs of borrowing funds from Islamic banking are lower than conventional financing (BF4) | 3.96 | 1.093 | -0.761 | -0.248 | 2.605 | 0.866 |
| Security | | | | | | |
| Islamic bank ensure all their operating system are updated with high security patches (SC1) | 4.22 | 1.045 | -1.242 | 0.710 | 2.822 | 0.890 |
| Islamic bank provides latest technology to stop unauthorized users (SC2) | 4.23 | 1.087 | -1.346 | 1.002 | 2.963 | 0.899 |
| I feel safe when I release my credit card information through internet banking (SC3) | 4.10 | 1.126 | -1.129 | 0.350 | 1.987 | 0.812 |
| The matter of high security of Islami bank influences me in using Islamic banking services (SC4) | 4.19 | 1.082 | -1.229 | 0.684 | 2.795 | |
| Perception | | | | | | |
| I am likely to choose Islamic banking (Halal) products (CP1) | 4.22 | 1.076 | -1.376 | 1.233 | 2.415 | 0.898 |
| Islamic banks serve the welfare of society (CP2). | 4.25 | 1.052 | -1.425 | 1.416 | 2.844 | 0.911 |
| Islamic bank is professional in online banking (CP3) | 4.29 | 1.029 | -1.544 | 1.839 | 2.573 | 0.926 |
| Islamic bank always fulfil their promises (CP4) | 4.28 | 1.054 | -1.539 | 1.749 | 2.154 | 0.917 |
| Satisfaction | | | | | | |
| I am satisfied with my bank's online services (SAT1) | 4.10 | 1.023 | -0.946 | 0.227 | 2.366 | 0.858 |
| I am very satisfied with my bank's online services (SAT2) | 4.14 | 1.025 | -1.098 | 0.593 | 2.551 | 0.876 |
| Assuming your entire experience with the Islamic Banks, I am satisfied (SAT3) | 4.05 | 1.192 | -0.989 | -0.216 | 1.689 | 0.811 |
| Islamic Banks exceed my expectations while offering quality services (SAT4) | 3.93 | 1.116 | -0.762 | -0.273 | 2.066 | 0.833 |
| Word of mouth | | | | | | |
| I will recommend my bank to other people (WOM1) | 4.17 | 1.004 | -1.077 | 0.582 | 2.032 | 0.862 |
| I would recommend my bank's website to others (WOM2) | 4.11 | 1.060 | -1.098 | 0.640 | 2.353 | 0.878 |
| I intend to continue using my bank online services (WOM3) | 4.19 | 1.040 | -1.246 | 0.970 | 2.155 | 0.879 |
| I prefer my bank above others (WOM4) | 4.20 | 1.008 | -1.204 | 0.959 | 2.969 | 0.870 |
| I recommend my family, friends and relatives to visit the Islamic Bank that I am already dealing with (WO5) | 4.08 | 1.069 | -0.958 | 0.173 | 2.723 | 0.856 |
| I will spread positive word of mouth about my Islamic Bank and its high quality of services (WOM6) | 4.22 | 1.017 | -1.206 | 0.827 | 2.739 | 0.858 |

Note: Standard deviation (SD), Collinearity statistics/ variance inflation factor (VIF), Factor loading (FL).

significant. Ethical responsibility and religious values have also a significant impact on the perception of Islamic banking services. The results are reported in Table 6.

## Discussion

Following the findings of this study, it is implicit that customers' perceptions regarding ethical responsibility, religious values, benefits, and security can increase customer satisfaction, which in turn promotes customers' word of mouth with others about Islamic banking services.

**Table 3. Convergent validity.**

| Characteristics | CA | rho_A | CR | AVE |
|---|---|---|---|---|
| Ethical responsibility | 0.859 | 0.861 | 0.904 | 0.703 |
| Religious value | 0.842 | 0.842 | 0.894 | 0.679 |
| Benefit | 0.860 | 0.869 | 0.905 | 0.704 |
| Security | 0.897 | 0.903 | 0.928 | 0.764 |
| Customer perceptions | 0.944 | 0.945 | 0.957 | 0.817 |
| Satisfaction | 0.866 | 0.869 | 0.909 | 0.714 |
| Word of mouth | 0.934 | 0.934 | 0.948 | 0.752 |

Note: Cronbach's alpha (CA), Composite reliability (CA), Average variance extracted (AVE).

Hypothesis 1 (H1) and hypothesis 2 (H2) indicated that there is a positive and significant relationship between ethical responsibility, religious value, and customer perceptions. Customers' perceptions of ethical responsibility in Islamic banking services can create customer satisfaction, which leads to customers' word of mouth (WOM) with others on the organizations' ethical commitment. Similar opinion were found with regards to the effect of ethical responsibility and religious value on customers' perceptions [11, 61, 62] in the area of health services and Islamic banking services in Jordan.

Hypothesis 3 (H3) revealed a negative significant relationship between benefits and customers' perception. This might be due to the difference in cultural and religious sentiment among different customers, for example Muslims and non-Muslims. Benefits for Islamic banks in Bangladesh refer to Islamic bank's profit, credit at favorable terms and conditions, lower service charges, and cost of borrowing funds from Islamic banks that are lower than conventional banks. This study found that there is a negative significant link between benefits and perceptions. Ahmad et al. [63] indicated that a negative confirmation result occurs when customer perceptions are poorer than their expectations, needs, desire, and appetite. Accordingly, many beneficial practices in Islamic banks may signal different information to the different backgrounds of customers of Islamic banks in Bangladesh, as both Muslim and non-Muslim

**Table 4. Discriminant validity.**

| Characteristics | BF | CP | ER | RV | SAT | SC | WOM |
|---|---|---|---|---|---|---|---|
| Fornell-Larcker Criterion | | | | | | | |
| Benefit (BF) | 0.839 | | | | | | |
| Customer perceptions (CP) | 0.606 | 0.904 | | | | | |
| Ethical responsibility (ER) | 0.764 | 0.742 | 0.838 | | | | |
| Religious value (RV) | 0.758 | 0.708 | 0.849 | 0.824 | | | |
| Satisfaction (SAT) | 0.713 | 0.809 | 0.762 | 0.718 | 0.845 | | |
| Security (SC) | 0.717 | 0.789 | 0.795 | 0.755 | 0.762 | 0.874 | |
| Word of mouth (WOM) | 0.664 | 0.803 | 0.727 | 0.696 | 0.785 | 0.745 | 0.867 |
| Heterotrait-Monotrait Ratio | | | | | | | |
| Benefit (BF) | | | | | | | |
| Customer perceptions (CP) | 0.665 | | | | | | |
| Ethical responsibility (ER) | 0.815 | 0.821 | | | | | |
| Religious value (RV) | 0.817 | 0.791 | 0.698 | | | | |
| Satisfaction (SAT) | 0.823 | 0.820 | 0.818 | 0.841 | | | |
| Security (SC) | 0.812 | 0.854 | 0.842 | 0.837 | 0.821 | | |
| Word of mouth (WOM) | 0.737 | 0.855 | 0.811 | 0.786 | 0.870 | 0.813 | |

**Table 5. Cross loadings.**

| Items | BF | CP | ER | RV | SAT | SC | WOM |
|---|---|---|---|---|---|---|---|
| Item BF1 | *0.790* | 0.466 | 0.579 | 0.583 | 0.510 | 0.547 | 0.499 |
| Item BF2 | *0.840* | 0.597 | 0.690 | 0.692 | 0.639 | 0.674 | 0.596 |
| Item BF3 | *0.857* | 0.463 | 0.637 | 0.614 | 0.602 | 0.563 | 0.544 |
| Item BF4 | *0.866* | 0.485 | 0.643 | 0.639 | 0.627 | 0.602 | 0.577 |
| Item CP1 | 0.557 | *0.898* | 0.707 | 0.644 | 0.737 | 0.740 | 0.733 |
| Item CP2 | 0.564 | *0.911* | 0.662 | 0.638 | 0.794 | 0.713 | 0.712 |
| Item CP3 | 0.551 | *0.926* | 0.677 | 0.652 | 0.722 | 0.703 | 0.741 |
| Item CP4 | 0.521 | *0.917* | 0.657 | 0.645 | 0.725 | 0.704 | 0.712 |
| Item ER1 | 0.634 | 0.601 | *0.841* | 0.748 | 0.664 | 0.648 | 0.606 |
| Item ER2 | 0.541 | 0.641 | *0.814* | 0.671 | 0.571 | 0.650 | 0.584 |
| Item ER3 | 0.697 | 0.575 | *0.827* | 0.675 | 0.652 | 0.644 | 0.585 |
| Item ER4 | 0.695 | 0.663 | *0.871* | 0.750 | 0.673 | 0.719 | 0.660 |
| Item RV1 | 0.678 | 0.567 | 0.705 | *0.832* | 0.621 | 0.638 | 0.587 |
| Item RV2 | 0.522 | 0.619 | 0.627 | *0.768* | 0.533 | 0.606 | 0.503 |
| Item RV3 | 0.655 | 0.544 | 0.730 | *0.838* | 0.612 | 0.615 | 0.589 |
| Item SAT1 | 0.581 | 0.646 | 0.646 | 0.592 | *0.858* | 0.634 | 0.641 |
| Item SAT2 | 0.628 | 0.667 | 0.675 | 0.640 | *0.876* | 0.643 | 0.714 |
| Item SAT3 | 0.549 | 0.810 | 0.624 | 0.587 | *0.811* | 0.687 | 0.659 |
| Item SAT4 | 0.656 | 0.587 | 0.628 | 0.603 | *0.833* | 0.601 | 0.630 |
| Item SC1 | 0.657 | 0.712 | 0.748 | 0.703 | 0.708 | *0.890* | 0.685 |
| Item SC2 | 0.615 | 0.741 | 0.721 | 0.688 | 0.681 | *0.899* | 0.645 |
| Item SC3 | 0.622 | 0.594 | 0.615 | 0.596 | 0.624 | *0.812* | 0.584 |
| Item SC4 | 0.619 | 0.699 | 0.687 | 0.647 | 0.650 | *0.893* | 0.686 |
| Item WOM1 | 0.586 | 0.720 | 0.606 | 0.604 | 0.684 | 0.649 | *0.862* |
| Item WOM2 | 0.598 | 0.693 | 0.639 | 0.627 | 0.691 | 0.646 | *0.878* |
| Item WOM3 | 0.539 | 0.691 | 0.649 | 0.582 | 0.688 | 0.622 | *0.879* |
| Item WOM4 | 0.551 | 0.691 | 0.613 | 0.595 | 0.676 | 0.665 | *0.870* |
| Item WOM5 | 0.567 | 0.655 | 0.614 | 0.572 | 0.660 | 0.607 | *0.856* |
| Item WOM6 | 0.616 | 0.726 | 0.664 | 0.642 | 0.685 | 0.689 | *0.858* |

customers are receiving Islamic banking products and services in this country. Muslims and non-Muslims have a multifaceted demand and all customers not only contain benefits but also other general and Islamic-related products and services, such as ethical responsibility, security, and religious value.

**Table 6. Path coefficient.**

| Hyp | Relationship | β | Sd | t-values | $f^2$ | $R^2$ | $q^2$ | Comment |
|---|---|---|---|---|---|---|---|---|
| H1 | ER -> CP | 0.248 | 0.094 | 2.644** | 0.239 | | | Significant |
| H2 | RV -> CP | 0.162 | 0.065 | 2.483* | 0.519 | | | Significant |
| H3 | BF -> CP | -0.088 | 0.051 | 1.741* | 0.038 | | | Significant |
| H4 | SC -> CP | 0.532 | 0.069 | 7.690** | 0.278 | 0.665 | 0.536 | Significant |
| H5 | CP -> SAT | 0.809 | 0.022 | 36.331** | 0.901 | 0.655 | 0.452 | Significant |
| H6 | SAT -> WOM | 0.785 | 0.032 | 24.407** | 0.603 | 0.616 | 0.458 | Significant |

Note: t-value ≥ 2.326 considers **$p<0.01$ and t-value ≥ 1.645 consider at *$p<0.05$.

Islamic banks in Bangladesh offer credit at favorable terms and conditions. Cost of borrowing funds from Islamic banking is also comparatively lower than conventional banking. Although most people would choose Islamic banks for their transactions, perhaps they might have different opinions when it comes to religious practices that anchor these products. As a result, benefits become negatively associated with the perception of Islamic banking services. Although this study identified a direct negative effect between benefits and customers' perception, Thambiah et al. [64] identified a significant positive impact of additional benefits that comes as a package with Islamic home loans in the context of Malaysia. Oladapo et al. [65] and Saqib et al. [66] contended that benefits have a significant impact on perception in the context of Shariah compliance in comparing the Islamic banking sector of Pakistan, Malaysia and Saudi Arabia.

In terms of hypothesis 4 (H4), security has a highly significant impact on customers' perception of Islamic banking services. This finding is relevant to the study conducted by Naeem [67] (2020) which focused on conventional and Islamic banking services. Naeem [67] identified customer perception of security and privacy to be reflective towards customers' satisfaction. Islamic banks endure updated operating systems, such as biometric authentication technology with high-security patches. Hence, the banks' commitment to strive towards high security measures influence customers in using Islamic banking services. Mansour et al. [36] stated that information technology security for Islamic banks is more developed than conventional banks. Currently, Islamic banking providers are using high-security technology to ensure the safety and security of banking transactions, which in turn reflect customers' satisfaction. Raza et al. [68] indicated the introduction of more sophisticated operating tools to evaluate the practicality of biometric verification innovation in web banking transaction services. Bank customers' perceptions of security related to the effectiveness of biometrics authentication innovation can increase customer satisfaction in terms of banking transactions and deposits and high returns to customers [69]. The importance of biometric technology in internet banking services can reflect customers' perceived privacy and security towards satisfaction with Islamic banking services.

Customers' perceptions are highly correlated with the satisfaction of the customer. Thus, hypothesis 5 (H5) is accepted. Customers are more likely to choose Islamic banking products and services, as Islamic banks serve the welfare of society and fulfill their promises. Islamic banking performs the roles of a commercial and investment bank, an investment trust, as well as an institution that manages investments, which are welfare for society. Islamic banking products are superior to traditional banking products in various ways. For instance, customers can determine how much they will pay the bank until the financing is complete, and Islamic banks charge a fixed amount for all their financial products. In essence, there is no financial risk. Thus, Islamic banking is becoming a more common practice as more banks use its products. It is no longer only a theoretical idealism; it is a fact of reality that influences customer perceptions of Islamic banking transactions. Findings from this study implies that higher perceptions regarding Islamic banking products and services would lead to an increase in the higher satisfaction of the customer. The outcome of this result is relevant to the studies of Alam and Al-amri [70], also Fida et al. [5], which highlighted the Islamic banking service and its impact on customer satisfaction.

Hypothesis 6 (H6) is accepted because there is a positive and highly significant relationship between satisfaction and WOM. This finding supports the study of Jridi et al. (2018), who examined customer satisfaction and WOM. Jridi et al. [71] found that customer satisfaction can reflect WOM with others for receiving services. Findings of this study hence imply that higher satisfaction can reflect higher WOM with others about Islamic banking services. Customers of Islamic banks are more satisfied with Islamic banks as they ensure the security,

safety, religious value, and ethical commitment, and also provide benefits with financial transactions. For example, lower service charges and lower monthly repayment of Islamic banks as compared to conventional banks. Hoque et al. [11], and Islam et al. [72] highlighted Islamic banking products and services as well as customer satisfaction. Marcos and Coelho [73] focused on holistic determinants of customer loyalty and WOM about insurance services. If customers are satisfied with their particular products and services, they will willingly recommend others.

## Theoretical and managerial implication

This study provides significant theoretical and managerial contributions. Theoretically, this study enriches the relevant literature on how ethical responsibility, religious value, benefit, and security affect the perceptions of Islamic banking services, which in turn reflect customer satisfaction and word of mouth (WOM) with others for transactions with Islamic banks in Bangladesh. The main motivation of this study is the ethical responsibility and security issues that Islamic banks should practice to ensure customer perception and satisfaction. Customers' perceptions of ethical practices of Islamic banks and security can lead to an increase in customers' satisfaction, which in turn reflects customers' WOM with others to encourage them to make transactions using Islamic banking products and services. Contribution from this study is not only significant for readers, but also for managers and practitioners. Practically, this study is crucial for managers and policymakers of Islamic banking institutions to consider the existing factors of this study that affect customers' perceptions of Islamic banking services, which reflect customer satisfaction and explain WOM for a large part of customer selection of Islamic banking services.

Findings of this study can provide managers of Islamic banks with better understanding on how to attain competitive advantage and create value for promoting bank customers in developing customers' perceptions, satisfaction, and WOM to others. This study identified four main factors of customers' perceptions, in which Islamic banks can improve particular products and services strategies. Islamic banks should allocate more resources to attract customers about their perceptions of products and services. The findings highlighted that customers' higher satisfaction reflects WOM about Islamic banking services. Islamic banks need to increase their efforts in educating customers about the benefits of Islamic banking products and services. Islamic banks can incorporate existing findings of this study into their management practices for customer satisfaction and WOM. Understanding and implementing the factors that reflect satisfaction and WOM of Islamic banking services would support managers in evolving and building long-term relationships with customers.

## Limitations and future direction

This study has several limitations despite its significant theoretical and practical contributions. Some limitations need to be considered particularly for the generalization of data collection method and results, which was conducted in a specific area. Future studies could be interesting with different types of banks over some geographic areas to generalize the findings and compare them between countries and banks in which promotion of Islamic banks is emerging. The main limitation of this study is reflected on data collection from Islamic banks which is mainly focused in Dhaka city. Thus, results from this study may not be representative of the entire population, and they may only be suitable for Bangladesh and other relevant backgrounds. Future studies need to use determinants of Islamic banks and Shariah-compliant issues in different contexts to identify potential explanatory factors for the involvement with Islamic banking services. This study hence proposes to integrate other relevant variables into

the model, such as awareness of Islamic finance, shariah-compliant attributes, perceived credibility, and commitment to Islamic banks.

## Supporting information

**S1 Dataset.**
(RAR)

## Author Contributions

**Conceptualization:** Muhammad Khalilur Rahman, Mohd Nor Hakimin Bin Yusoff.

**Data curation:** Farhana Begum.

**Formal analysis:** Muhammad Khalilur Rahman.

**Funding acquisition:** Sharifah Norzehan Syed Yusuf.

**Investigation:** Sharifah Norzehan Syed Yusuf.

**Methodology:** Muhammad Khalilur Rahman.

**Resources:** Muhammad Nazmul Hoque, Sharifah Norzehan Syed Yusuf, Farhana Begum.

**Software:** Muhammad Khalilur Rahman.

**Supervision:** Muhammad Khalilur Rahman, Muhammad Nazmul Hoque.

**Validation:** Mohd Nor Hakimin Bin Yusoff.

**Visualization:** Sharifah Norzehan Syed Yusuf, Mohd Nor Hakimin Bin Yusoff.

**Writing – original draft:** Muhammad Khalilur Rahman, Muhammad Nazmul Hoque.

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
