## [Decision Letter · Decision Letter 0]

8 Sep 2022

PONE-D-22-13158Do customers’ perceptions of Islamic banking services reflect word of mouth? Evidence from Islamic banks in BangladeshPLOS ONE

Dear MUHAMMAD KHALILUR RAHMAN,

Thank you for submitting your manuscript to PLOS ONE. After careful consideration, we feel that it has merit but does not fully meet PLOS ONE’s publication criteria as it currently stands. Therefore, we invite you to submit a revised version of the manuscript that addresses the points raised during the review process.

We look forward to receiving your revised manuscript.

Kind regards,

Li-Xin Zhong, Ph.D.

Academic Editor

PLOS ONE

Journal Requirements:

2. LOS ONE does not copy edit accepted manuscripts (https://journals.plos.org/plosone/s/criteria-for-publication#loc-5). To that effect, please ensure that your submission is free of typos and grammatical errors.

The authors have declared that no competing interests exist.

Powered

Reviewers' comments:

Reviewer's Responses to Questions

**Comments to the Author**

1. Is the manuscript technically sound, and do the data support the conclusions?

Reviewer #1: Yes

Reviewer #2: Yes

2. Has the statistical analysis been performed appropriately and rigorously? 

Reviewer #1: Yes

Reviewer #2: Yes

3. Have the authors made all data underlying the findings in their manuscript fully available?

Reviewer #1: Yes

Reviewer #2: Yes

4. Is the manuscript presented in an intelligible fashion and written in standard English?

Reviewer #1: Yes

Reviewer #2: Yes

5. Review Comments to the Author

Reviewer #1: This paper needs clarity of some factors like Benefit, Security, Words of mouth as well as sequence in the text and correct use of terminology. The explanation of these factor is ambiguous to some an extent. Detail review is attached for necessary guidance.

Reviewer #2: The paper investigates the customers’ perceptions of Islamic banking services and their impact on satisfaction and word of mouth with others. The results show that ethical responsibility, religious value, benefits and security affect the customers’ perceptions, the customers’ perceptions effect customers’ satisfaction and customers’ satisfaction affects customers’ perception and word of mouth. Therefore, the customers’ perception and satisfaction are the intermediate variable of Islamic banking services and word of mouth. Although this finding is more comprehensive than previous studies and the data and research perspective is novel, I think that the paper needs a substantial revision before assessing whether it should be considered for publication. I give detailed comments below.

1. The structure of the paper is a little confusing, especially the Introduction, which does not highlight what problems are raised, what problems are found, and what problems are solved.

2. Some results, such as the relationship between security and customers’ perception, and the relationships between religious values and customers’ perception of Islamic banking services were confirmed by previous studies. Does this paper only establish a comprehensive impact transmission mechanism which is shown in Figure 1 based on previous studies? What is the innovation of this paper compared with the previous studies?

3. Assumptions of model in this paper need to be modified. The title of the article is do customers' perceptions of Islamic banking services reflect word of mouth? There are two dependent variables, one is customers' perception of Islamic banking services, and the other is word of mouth rewards Islamic banking services. And only one hypothesis involves the influence of variables on word of mouth. Do you need to think again? In the assumptions of model, author did not explain whether the variables are expected to be positively or negatively correlated. It is better to explain it and then discuss it.

4. This paper proposes unidirectional causality among customers’ perceptions, customers’ satisfaction and word of mouth. Whether the customers’ satisfaction will improve the customer's perception of Islamic banking services is a question worthy of further discussion.

5. In the part "Data collection procedure", you said "Before data collection, we also conducted a pilot test study with 30 samples for the validity of the questionnaire." How did you test the validity? How was the validity of the final questionnaire?

6. In Table 1, you give the details of the respondents. Table 1 shows the differences in gender, income and age among respondents. Will these differences affect results in this paper? For example, there may be differences in perception between male and female.

7. Mediating effect can be deleted from this section.

8. The logic of the discussion part is also confusing, and there is no clear explanation

6. PLOS authors have the option to publish the peer review history of their article (what does this mean?). If published, this will include your full peer review and any attached files.

Reviewer #1: **Yes: **Prof.Dr.Abdul Ghafoor Awan

Reviewer #2: **Yes: **Aizhong Shen

---

## [Author Response · Author response to Decision Letter 0]

28 Sep 2022

Response to reviewers’ comment

Manuscript ID: PONE-D-22-13158

Title: Do customers’ perceptions of Islamic banking services reflect word of mouth? Evidence from Islamic banks in Bangladesh

Journal name: PLOS ONE

Dear Editor,

Thank you for these comments designed to improve our paper. We have addressed below the response to the reviewers’ comments. We greatly appreciate the time and effort put forth by reviewers and editors to improve our paper. If any responses are unclear or you wish additional changes, please let us know. All the changes are made using “track changes”. Also, for better tracking, the addressed comments from each reviewer are blue colour coded into the manuscript. 

Review Comments to the Author

Reviewer #1: This paper needs clarity of some factors like Benefit, Security, Words of mouth as well as sequence in the text and correct use of terminology. The explanation of these factor is ambiguous to some an extent. Detail review is attached for necessary guidance.

Author’s Response: Thank you for your nice comment. We have made the necessary amendment to the benefits, security and word of mouth in the literature review section. Please see pages no 4-6, for benefits, refer to line no. 189-199, for security, line no. 213-234, and for word of mouth, line no. 256-260 (blue colour text). 

Comment 1. I reviewed the above paper and submit the following suggestions for improvement of its quality: -

One of the main variables in this study is “words of mouth” which the authors have analyzed and tried to prove effective for future growth of Islamic banking services. But in the model it is shown as dependent variable. The authors must know that only independent variables have direct while mediating or moderating variables have indirect effect on dependent variable, which is always one. If the author wants to measure the impact of “Words of Mouth” on the satisfaction of customers and banking services, they should use it as an independent variable and measures its impact on expansion of Islamic banking.

Author’s Response: Thank you very much for your nice comment. We have revised the model and deleted the mediating effect based on the reviewers’ comments and suggestions. In this study, we have measured the word of mouth through bank customers’ satisfaction and their perceptions of Islamic banking services. Hence, word of mouth is a form of loyalty, thus, if the customers are satisfied, they would likely recommend others for Islamic banking services. Please see page no. 6, figure 1. 

Comment 2. According to authors, “security has a highly significant impact on customer perception of Islamic banking services, because Islamic banks endure updated operating systems with high-security patches and the matter of this high-security influences customers in using Islamic banking services.” 

The author should clearly illustrate what type of distinct “operating system” Islamic banks has and how it is different from conventional banks as well as how it ensures the deposit, transactions and high return to customers.

Author’s Response: Thank you professor for your nice comment. We have made the amendment to the security and operating system. Please see page no. 13, line no. 433-445 (blue colour text). 

Comment 3. The author stated that “the fifth author of this study physically distributed 700 questionnaires to the customers of Islamic banks in Bangladesh between the time period of April and July 2021. We received 385 responses from the bank customers out of which we identified a total of 377 valid responses for data, 53.86% response rate of data collection.” 

The author could not justify why the response rate was so much low when 700 questionnaires were physically distributed among respondents. The author must disclose why 315 questionnaires were not included into the analysis. Mostly the response rate is more than 80 percent when data is collected through survey method as the author has done. Here response rate is very low. It must be justified through valid arguments.

Author’s Response: Thank you for your comment. We have made the amendment for the low response rate of this survey. Please see page no. 8, line no. 330-337 (blue colour text). 

Comment 4. Capital R-square (R2 ) is used to show goodness of fit of the model and ratio of variation in the dependent variable on account of independent variables. But author wrote r-square to denote the model, which is not correct. The must understand difference between r-square and R-square.

Author’s Response: Thank you very much for your comment and suggestion. We have made the correction. Please see page no. 12, line no. 332, and table no. 6 (blue colour text). 

Comment 5. The author could not explain the meaning of another important factor “Benefit” properly. They must have explained what is its meaning in Islamic banking and what types of benefits Islamic banks offer to their customers. Moreover, the empirical results also show negative association between benefit and perception of Islamic banking services. When the customers have negative perception about the benefit to be accrued from Islamic banks how can they refer or convince others to opt Islamic banking services through the “word of mouth”.

The authors have also failed to justify why all other studies such as Thambiah et al. (2011), Oladapo et al. (2021) and Saqib et al. (2016) conducted in Pakistan, Malaysia and Saudi Arabia have positive association between benefit and perception of Islamic banking services while their study has negative association between two variables. 

Author’s Response: Thank you very much for your nice comment. We have made the amendment. Please see page no. 13, line no. 454-466 (blue colour text). 

Comment 6. The authors states that “the customers choose Islamic banking products and services as Islamic banks served the welfare of society and fulfilled their promises”.

They must have defined welfare role or services and promises of Islamic Banks because all banks whether Islamic or conventional are commercial business organization and their prime objective is to maximize their profit. Conventional Banks also fulfill their promises by giving fixed return to their depositors while Islamic banks do not make any promise about fixed return because they collect all deposits under profit and loss sharing mechanism. The author must illustrate this point for satisfaction of the customers of Islamic Banks and general readers of this paper.

Author’s Response: Thank you very much for your comment. We have made the amendment. Please see page no. 14, line no. 446-485 (blue colour text). 

7. There are grammatical errors which may be removed through proper editing by an English expert.

Author’s Response: Thank you. Based on the comments and suggestions, we have edited and proofread the manuscript by a professional proofreader. 

Reviewer #2: The paper investigates the customers’ perceptions of Islamic banking services and their impact on satisfaction and word of mouth with others. The results show that ethical responsibility, religious value, benefits and security affect the customers’ perceptions, the customers’ perceptions effect customers’ satisfaction and customers’ satisfaction affects customers’ perception and word of mouth. Therefore, the customers’ perception and satisfaction are the intermediate variable of Islamic banking services and word of mouth. Although this finding is more comprehensive than previous studies and the data and research perspective is novel, I think that the paper needs a substantial revision before assessing whether it should be considered for publication. I give detailed comments below.

Comment 1. The structure of the paper is a little confusing, especially the Introduction, which does not highlight what problems are raised, what problems are found, and what problems are solved.

Author’s Response: Thank you very much for your nice comment. Based on the comment, we have made the necessary amendment to the introduction section. Please see page no. 1, line no. 54-67 (blue colour text). 

Comment 2. Some results, such as the relationship between security and customers’ perception, and the relationships between religious values and customers’ perception of Islamic banking services were confirmed by previous studies. Does this paper only establish a comprehensive impact transmission mechanism which is shown in Figure 1 based on previous studies? What is the innovation of this paper compared with the previous studies?

Author’s Response: Thank you very much for your nice comment. Based on the review of the literature, this study establishes the conceptual model in Figure 1. The main motivation of this study is the ethical responsibility and security issues that Islamic banks should practice to ensure customer perception and satisfaction, which is a different and new innovation compared with the previous studies in the context of Islamic banks in Bangladesh. Customer perceptions of ethical practices of Islamic banks and security can lead to increase customer satisfaction, which in turn reflects customer word of mouth with others to encourage them to receive Islamic banking products and services. Please see page no. 14, line no. 488-492 (blue colour text). 

Comment 3. Assumptions of model in this paper need to be modified. The title of the article is do customers' perceptions of Islamic banking services reflect word of mouth? There are two dependent variables, one is customers' perception of Islamic banking services, and the other is word of mouth rewards Islamic banking services. And only one hypothesis involves the influence of variables on word of mouth. Do you need to think again? In the assumptions of model, author did not explain whether the variables are expected to be positively or negatively correlated. It is better to explain it and then discuss it. 

Author’s Response: Thank you very much for your nice comment. We have made the necessary amendment to the negative and positive correlation of the variables. According to the comment, we have revised the model in figure 1, and we have deleted the mediating variable. In this study, we have examined the word of mouth through customer perceptions (e.g., ethical responsibility, religious value, benefit, and security), and customer satisfaction. According to the findings in Table 6, we highlighted that customer perceptions can reflect satisfaction, which in turn encourages customer word of mouth with others. We have explained the positive and negative correlation of the variables. Please see page no. 12 in the ‘structural model analysis section’, line no. 389-400, and, discussion section page no. 12-13, line no. 417-431, and 445-460 (blue colour text).

Comment 4. This paper proposes unidirectional causality among customers’ perceptions, customers’ satisfaction and word of mouth. Whether the customers’ satisfaction will improve the customer's perception of Islamic banking services is a question worthy of further discussion.

Author’s Response: Thank you very much for your comment. In this study, based on the proposed mode in figure 1, we have examined that customer perceptions including ethical responsibility, religious value, benefits, and security will increase customer satisfaction, which in turn leads to customer word of mouth about the Islamic banking services. The results revealed that customer perceptions will improve customer satisfaction and in turn, it will increase word of mouth with others. For details results and discussion, please see page no. 12, table 6, and discussion section, page no. 12-13 (blue colour text). 

Comment 5. In the part "Data collection procedure", you said "Before data collection, we also conducted a pilot test study with 30 samples for the validity of the questionnaire." How did you test the validity? How was the validity of the final questionnaire?

Author’s Response: Thank you very much for your nice comment. Based on the comment, we have made the amendment. Please see page no. 7-8, line no. 297-314. For the reliability and validity of the final questionnaire, please see page no. 12, the findings in Table no 6. 

Comment 6. In Table 1, you give the details of the respondents. Table 1 shows the differences in gender, income and age among respondents. Will these differences affect results in this paper? For example, there may be differences in perception between male and female.

Author’s Response: Thank you very much for your comment. The percentage of the gender, age, and income level of the respondents are different because it is estimated based on the respondent’s participation in the study. The target of the respondents was purposively selected to be bank customers but questionnaires were distributed randomly to the respondents. Thus, these differences are based on the participation of the respondents, particularly male and female respondents in this study, these differences will not affect the results of this study.

Comment 7. Mediating effect can be deleted from this section.

Author’s Response: Thank you. We have deleted the mediating effect. Please see page no. 7, in figure 1. 

Comment 8. The logic of the discussion part is also confusing, and there is no clear explanation

Author’s Response: Thank you very much. Based on the comment, we have made the amendment. Please see page no. 12-14 (blue colour text).

---

## [Decision Letter · Decision Letter 1]

20 Dec 2022

Do customers’ perceptions of Islamic banking services predict satisfaction and word of mouth? Evidence from Islamic banks in Bangladesh

PONE-D-22-13158R1

Dear Dr. Hoque,

We’re pleased to inform you that your manuscript has been judged scientifically suitable for publication and will be formally accepted for publication once it meets all outstanding technical requirements.

Kind regards,

Li-Xin Zhong, Ph.D.

Academic Editor

PLOS ONE

Additional Editor Comments (optional):

Reviewers' comments:

Reviewer's Responses to Questions

**Comments to the Author**

1. If the authors have adequately addressed your comments raised in a previous round of review and you feel that this manuscript is now acceptable for publication, you may indicate that here to bypass the “Comments to the Author” section, enter your conflict of interest statement in the “Confidential to Editor” section, and submit your "Accept" recommendation.

Reviewer #1: All comments have been addressed

Reviewer #2: All comments have been addressed

2. Is the manuscript technically sound, and do the data support the conclusions?

Reviewer #1: Yes

Reviewer #2: Yes

3. Has the statistical analysis been performed appropriately and rigorously? 

Reviewer #1: Yes

Reviewer #2: Yes

4. Have the authors made all data underlying the findings in their manuscript fully available?

Reviewer #1: Yes

Reviewer #2: Yes

5. Is the manuscript presented in an intelligible fashion and written in standard English?

Reviewer #1: Yes

Reviewer #2: Yes

6. Review Comments to the Author

Reviewer #1: Some minor flaws were found in the revised paper, and I have mentioned them in my second review. The author may be asked to incorporate them.

Regards

Reviewer #2: (No Response)

7. PLOS authors have the option to publish the peer review history of their article (what does this mean?). If published, this will include your full peer review and any attached files.

Reviewer #1: **Yes: **Prof.Dr.Abdul Ghafoor Awan

Reviewer #2: No

---

## [Editor Report · Acceptance letter]

27 Dec 2022

PONE-D-22-13158R1 

Do customers’ perceptions of Islamic banking services predict satisfaction and word of mouth? Evidence from Islamic banks in Bangladesh 

Dear Dr. Hoque:

I'm pleased to inform you that your manuscript has been deemed suitable for publication in PLOS ONE. Congratulations! Your manuscript is now with our production department. 

Kind regards, 

on behalf of

Dr. Li-Xin Zhong 

Academic Editor

PLOS ONE